# Is the Bacterial Cellulose Membrane Feasible for Osteopromotive Property?

**DOI:** 10.3390/membranes10090230

**Published:** 2020-09-12

**Authors:** Ana Paula Farnezi Bassi, Vinícius Ferreira Bizelli, Leticia Freitas de Mendes Brasil, Járede Carvalho Pereira, Hesham Mohammed Al-Sharani, Gustavo Antonio Correa Momesso, Leonardo P. Faverani, Flavia de Almeida Lucas

**Affiliations:** 1Department of Diagnosis and Surgery, São Paulo State University, UNESP, School of Dentistry, Araçatuba, São Paulo 16015-050, Brazil; viniciusbizelli@gmail.com (V.F.B.); leticiafmbr@gmail.com (L.F.d.M.B.); institutojarede@gmail.com (J.C.P.); gustavomomesso@gmail.com (G.A.C.M.); leonardo.faverani@unesp.br (L.P.F.); 2Department of Oral and Maxillofacial Surgery, College of Dentistry, Ibb University, Ibb 16015-050, Yemen; hishamm2010@live.com; 3Department of Animal Clinic, Surgery and Reproduction, São Paulo State University, UNESP, School of Veterinary Medicine, Araçatuba, São Paulo 16050-698, Brazil; flavia.lucas@unesp.br

**Keywords:** biomaterials, xenografts, cellulose

## Abstract

Guided bone regeneration was studied to establish protocols and develop new biomaterials that revealed satisfactory results. The present study aimed to comparatively evaluate the efficiency of the bacterial cellulose membrane (Nanoskin^®^) and collagen membrane Bio-Gide^®^ in the bone repair of 8-mm critical size defects in rat calvaria. Seventy-two adult male rats were divided into three experimental groups (n = 24): the CG—membrane-free control group (only blood clot, negative control), BG—porcine collagen membrane group (Bio-Guide^®^, positive control), and BC—bacterial cellulose membrane group (experimental group). The comparison periods were 7, 15, 30, and 60 days postoperatively. Histological, histometric, and immunohistochemical analyses were performed. The quantitative data were subjected to 2-way ANOVA and Tukey’s post-test, and *p* < 0.05 was considered significant. At 30 and 60 days postoperatively, the BG group showed more healing of the surgical wound than the other groups, with a high amount of newly formed bone (*p* < 0.001), while the BC group showed mature connective tissue filling the defect. The inflammatory cell count at postoperative days 7 and 15 was higher in the BC group than in the BG group (Tukey’s test, *p* = 0.006). At postoperative days 30 and 60, the area of new bone formed was greater in the BG group than in the other groups (*p* < 0.001). Immunohistochemical analysis showed moderate and intense immunolabeling of osteocalcin and osteopontin at postoperative day 60 in the BG and BC groups. Thus, despite the promising application of the BC membrane in soft-tissue repair, it did not induce bone repair in rat calvaria.

## 1. Introduction

Guided bone regeneration (GBR), a technique used to promote bone reformation, mainly depends on the use of a biocompatible membrane that acts as a physical barrier to prevent the adjacent connective tissue from invading the bone defect, thus creating a favorable space for bone regeneration [1]. During the healing process, the epithelial tissue migrates quickly to the wound, which complicates the process of bone regeneration [2].

These membranes must have the following properties: osteoinductivity, resorbability, biocompatibility, lack of cytotoxicity, and mechanical stability, that is, the capacity to maintain space during the process of bone repair [3,4].

Collagen-based membranes (CBMs) are well recognized for being biocompatible and hemostatic, promoting chemotaxis for fibroblasts and osteoblasts, and for being semipermeable, allowing the transfer of nutrients [5]. CBMs are also significant in the repair of intraosseous defects in the periodontium [6], and, when associated with several types of bone grafts, the efficacy of these membranes is improved, increasing their capacity to stimulate the repair of periodontal tissues [7,8]. Thus, this membrane is widely used in maxillofacial surgery and is considered the gold standard in bone-healing treatment. Among these collagen membranes, Bio-Gide^®^ (Bio-Gide^®^—Geistlich Wolhusen, Switzerland) stands out in the market.

We are currently looking for other options to improve bone tissue repair since, although very efficient, collagen membranes are expensive. As such, one of the alternatives is the exploitation of cellulose, which, besides presenting good biomechanical properties, is a natural polymer, biodegradable, and renewable. Therefore, it has become the subject of a series of investigations on tissue engineering [9,10,11,12].

One of the methods used to obtain cellulose is bacterial synthesis (bacterial cellulose), which results in cellulose with better properties than that extracted directly from plants, showing high mechanical strength, high water retention capacity [13], crystallinity, a high degree of biocompatibility, and resistance to degradation, making it suitable for use as a raw material in the production of membranes and scaffolds directed to a variety of tissues, including bone tissue [14,15,16]. In addition, this material has already been applied as an excellent substitute for skin in deep burn treatments [11,17], in treating extensive loss of the dermal and epidermal layers in chronic or acute wounds, in relieving pain and improving healing by accelerating the development of granulation tissue and new epithelium, and in reducing scar formation [11,16]. In this context, a cellulosic membrane produced by the action of some bacteria and yeast species on green tea (Nanoskin^®^) has been recently developed and is already being used as an aid in the repair of skin lesions, showing great results.

Cellulose membranes remain in the body for a long time without being degraded [11], but their contact with tissues does not generate toxicity or inflammation [11,17,18]. Bacterial cellulose is produced by strains of Gram-negative bacteria and is quite different from plant cellulose [19,20].

Tissue engineering studies present bacterial cellulose-containing biomaterials that are also efficient in the bone repair process, being explored mainly in the form of scaffolds [21,22]. The modification of the bacterial cellulose membrane with different scaffolds has also demonstrated promising results in in vivo osteoblast differentiation in calvaria defects [23]. 

Thus, this study aimed to compare the effectiveness of a bacterial cellulose membrane (Nanoskin^®^ Innovatec’s, São Carlos SP, Brazil) and a collagen membrane (Bio-Gide^®^—Geistlich Wolhusen, Switzerland) in the bone repair of male adult rat skulls (3 to 4 months) at postoperative 7, 15, 30, and 60 days, using the hypothesis that Nanoskin^®^ promotes bone formation compatible with that observed by Bio-Gide^®^.

## 2. Materials and Methods

### 2.1. Development of the Bacterial Cellulose Membrane

Nanoskin^®^ Innovatec’s bacterial cellulose raw material (São Carlos, São Paulo, Brazil) was supplied by Innovatec-Produtos Biotecnológicos Ltda. (São Carlos São Paulo, Brazil, FIESP: 80591940001, ISO: 13485:2003/AC:2009). The acetic fermentation process was performed using glucose as a source of carbohydrates and green tea (100%) as a source of nitrogen, which are natural and rich sources of polyphenols. The bacteria were then inoculated into the culture medium, and after being added, the medium was autoclaved at 100 °C. The results of this process were vinegar and nanobiocellulose biomass. Bacterial cellulose is produced by the Gram-negative bacterium *Gluconacetobacter xylinus* and extracted from the culture medium in a pure 3D structure, formed by an ultra-thin network of highly hydrated (3–8 nm) cellulose nanofibers (99% in weight), displaying high molecular weight, high cellulose crystallinity (60–90%), enormous mechanical strength, and full biocompatibility [18,20,22,23,24,25,26,27].

### 2.2. Samples

This study was submitted to and approved by the Ethics Committee of Araçatuba Dental School—UNESP under protocol number 2015-00965 and followed the ARRIVE Guidelines. 

Seventy-two male, adult (3 to 4 months) rats (Rattus novergicus albinus, Wistar), weighing approximately 200–300 g, were divided into three groups (n = 24 per group) and euthanized at four time points during the experiment: 7, 15, 30, and 60 days postoperatively. The sample size was calculated using the software SigmaPlot 12.0 (exact graphs and data analysis, Sant Jose, CA, USA). It was used as described in a previously published manuscript, in which the minimal difference in means of percentage bone fill was 18.9; expected standard deviation 8.1; for a power, test = 80%, and *p* > 0.005, it would be necessary to have five samples to an experimental group [28]. Thus, there was a possibility of animal loss during the research, and six animals per group were elected.

These animals were kept in cages, being four per cage identified with the code of each group and fed with balanced feed (NUVILAB, Curitiba PR, Brazil) containing 1.4% Calcium and 0.8% Phosphorus and water ad libitum in the Vivarium of the Araçatuba Dental School—UNESP. In each animal, a critical bone defect was created in the skull (8 mm), as described below:

**CG: clot group** (negative control)—n = 24: The critical bone defect was filled with blood clots; six rats were euthanized during each period of analysis (7, 15, 30, and 60 days postoperatively).

**BG: Bio-Gide^®^ group** (positive control)—n = 24: The critical bone defect was filled with blood clots, and a collagen membrane was placed over the defect; 6 rats were euthanized during each period of analysis (7, 15, 30, and 60 days postoperatively) (Figure 1A).

**BC: bacterial cellulose group** (experimental group)—n = 24: The critical bone defect was filled with blood clots, and a bacterial cellulose membrane was placed over the defect; 6 rats euthanized during each period of analysis (7, 15, 30, and 60 days postoperatively) (Figure 1B).

The division of experimental groups was performed through randomization using a lottery. An envelope was used with 24 papers containing the words CG, 24 containing the words BG, and 24 containing the words BC. Thus, 72 animals were divided aleatorily.

### 2.3. Experimental Surgery

The animals were subjected to an 8 h preoperative fast and were sedated by intramuscular administration of ketamine hydrochloride (Francotar—Vibrac do Brasil Ltda, São Paulo, Brazil) associated with xylazine (Rompum—Bayer AS—Health Animal, São Paulo, Brazil), at a dosage of 50 mg/kg and 5 mg/kg, respectively. A strict aseptic protocol was adopted, including sterilization of the instruments used, delimitation of the area to be operated with sterile fields, and use of sterile surgical gloves and gowns. All surgical procedures were performed in the Vivarium’s surgical room at the Araçatuba Dental School—UNESP. Trichotomy was then performed in the skull of the rats, followed by antisepsis with polyvinyl pyrrolidone iodine (10% PVPI, Riodeine Degermante, Rioquímica, São José do Rio Preto), associated with topical PVPI (10% PVPI, Riodeine, Rioquímica, São José do Rio Preto).

A V-shaped incision was made in the occipitofrontal direction, with the apex located in the frontal region, and the base located in the occipital region, measuring approximately 2 cm, with a detachment flap (Figure 2). Subsequently, using a 7-mm diameter inner drill bit (3i Implant Innovations, Inc., Palm Beach Gardens, FL, USA), coupled with low rotation under abundant irrigation with 0.9% sodium chloride solution (Darrow, Rio de Janeiro, Brazil), a critical surgical defect, 8 mm in diameter [21], was made in the central portion of the skull involving a sagittal suture, maintaining the integrity of the dura mater (Figure 2). In the CG (clot) group, the surgical critical-size defect was filled with a blood clot without overcoating of the defect. In the BG (Bio-Guide^®^) group, the surgical critical-size defect was filled with a blood clot and was covered with a porcine collagen membrane (Bio-Gide^®^; Geistlich Pharma AG, Wolhusen, Switzerland). In the BC (Bacterial cellulose^®^) group, the surgical critical-size defect was filled with a blood clot and covered by a bacterial cellulose membrane (Nanoskin^®^ Innovatec’s bacterial cellulose raw material, São Carlos SP, Brazil). 

At the end of the procedure, the soft tissues were carefully repositioned and sutured in planes using a resorbable suture (polylactic acid—Vycril 4.0, Ethicon, Johnson Prod., São José dos Campos, Brazil) in the deep plane and monofilament sutures (Nylon 5.0, Mononylon, Ethicon, Johnson Prod., São José dos Campos, Brazil) with interrupted sutures in the surface plane.

In the immediate postoperative period, each animal received a single intramuscular dose of 0.2 mL of penicillin G-benzathine (Pentabiótico Veterinário Pequeno Porte, Fort Dodge Saúde Animal Ltda., Campinas, SP, Brazil). Every two days, the cages were cleaned, and the animals cared.

### 2.4. Histological/Histometric Analysis

The histological blades were stained with hematoxylin and eosin (Merck & Co., Inc, Kenilworth, New Jersey, NY, USA). The histometric analysis was performed through the inflammatory cell and vessel count; the inflammatory response was determined, and the neoformed bone area was measured. The photomicrograph of the blades was made using an optical microscope (LeicaR DMLB, Heerbrugg, Switzerland) coupled to an image capturing camera (LeicaR DC 300F microsystems Ltd, Heerbrugg, Switzerland) and connected to a microcomputer with ImageJ digitized image analyzer software (National Institutes of Health, Bethesda, MD, USA).

All analyzed blades had their identification hidden so that the examiner was blind. For the inflammatory cell and vessel count, two sections per animal were analyzed, totalizing 48 sections, and three regions were evaluated: center of the defect, right side, and left side of the defect [12]. In the original objective of ×100, 130 points were predetermined, and the ones that touched a cell were counted. The area of bone tissue present in the central region of each bone defect was evaluated using two sections per animal (primary outcome). The data obtained in the analyses were transformed into absolute values, from pixels to square micrometers, to minimize the interference of the negative size difference. For comparison between the mean values obtained in the different experimental groups and periods, the data were subjected to initial statistical tests [12,29,30].

### 2.5. Immunohistochemical Analysis

Immunohistochemical reactions were visualized using the indirect immunoperoxidase detection method. Blockage of nonspecific reactions through the inactivation of endogenous peroxidase was performed using 3% hydrogen peroxide (Merck, São Paulo, SP, Brazil), 1% bovine serum albumin (Sigma-Aldrich Ltda., São Paulo, SP, Brazil), and 20% fat-free powdered milk. Antigen recovery was achieved with citrate phosphate buffer (pH 6.0) in the presence of humid heat. Primary antibodies against osteocalcin (OC) (Santa Cruz Biotechnology, Dallas, TX, USA) and osteopontin (OP) (Santa Cruz Biotechnology, Dallas, TX, USA) were used. These proteins were chosen to evaluate the cell responses related to bone mineralization. The secondary antibody used was a biotinylated anti-goat antibody produced in rabbits (Pierce Biotechnology, Rockford, IL USA), along with biotin and streptavidin (Dako, Glostrup, Denmark) and diaminobenzidine (Dako, Glostrup, Denmark). Counterstaining was performed using Harris hematoxylin.

All analyzed blades had their identification hidden so that the examiner was blind. The images were captured from the center of the defect performed in the skull of all the animals in the experiment for the biomarkers OC and OP. To acquire the images, a photomicroscope (LeicaR DMLB, Heerbrugg, Switzerland) connected to a microcomputer was used. The postoperative 15, 30, and 60-day periods alone were evaluated because at 7 days postoperatively, no cellular activity can be expected regarding the OC and OP biomarkers, which are related to the final stage of bone formation.

The immunohistochemical reactions were evaluated by assigning scores based on a semi-quantitative analysis [31,32,33,34,35,36,37]. The scores were assigned as null (0), mild (1), moderate (2), and intense (3). The increase in score represented an increased region of diaminobenzidine-stained cells. From 10 to 30% of stained cells were assigned score 1, from 50 to 70% of stained cells score 2, and from 80 to 100% or more of stained cells score 3. A calibrated evaluator performed the analysis of the images at two different times, at least 15 days between the analyses. Both accounted cells were tabulated and subjected to the Kappa test, obtaining an index > 0.9, which indicated concordance. 

### 2.6. Statistical Analysis

The data obtained through a histometric analysis were initially subjected to a homogeneity test to evaluate the distribution of data in a normal distribution curve (Shapiro–Wilk, *p* > 0.05). After the normality of the distribution was confirmed, two-way ANOVA and Tukey’s post hoc tests were used to compare the means. The level of significance was set at *p* < 0.05. For data from immunohistochemical analyses, the scores were subjected to variance analysis—two-way ANOVA test and Holm–Sisak post-test, considering the source of variation membranes and periods of analysis.

## 3. Results

As an exclusion criterion, it was established that any trans or postoperative surgical complication would exclude the sample from the evaluations. Among the samples, no complications were observed during the surgical procedure, and all animals went through a normal and healthy postoperative period, allowing the inclusion of everyone in the work.

### 3.1. Morphological Evaluation (Microscopic)

The results were evaluated using an optical microscope with a standardized reading of the slides of the CG, BC, and BG groups.

At day 7 postoperatively, all groups (CG, BG, and BC) showed hypervascularization, and the BC group exhibited an intense inflammatory infiltrate. On day 15, only the BG group presented osteoid tissue in the center of the defect, whereas the BC group showed a small inflammatory tissue that was still maintained. On day 30, the BG group had a large amount of bone tissue interspersed with fragments of porcine collagen membrane, and for the CG and BC groups, the defect could not be closed (Figure 3). On day 60, the CG defect was filled with connective tissue, without signs of bone neoformation, proving to be a critical defect; the BG group showed behavior as expected, promoting complete closure of the defect without the presence of membrane remnants. The BC group showed connective tissue, collagen fibers, and inflammatory infiltrate with discrete areas of osteoid tissue at the center of the defect (Figure 3 and Figure 4A,B).

### 3.2. Histometric Analysis

#### 3.2.1. Inflammatory Cells and Membrane 

Regarding the behavior of the membrane during the analysis periods, a statistical difference was noticed between the BC and BG groups (*p* < 0.05) in inflammatory cells and vessels. The cells noticed during the analysis were mostly lymphocytes and some monocytes, with a higher incidence in the BC group in all analysis periods. Concerning the postoperative periods of 7 and 15 days, between the BC and BG groups, there was a decrease in the number of inflammatory cells (*p* < 0.001), but an increase in the number of vessels in the BG group and a reduction in the BC group. When comparing the BG and BC groups, there was a discrepancy in the number of inflammatory cells at 7 days (*p* = 0.019) and no statistical difference at 15 days (*p* = 0.072). Regarding vessel count, at 7 days, similar values were observed between the groups (*p* = 0.163); however, at 15 days, a considerable statistical difference was observed, suggesting that the inflammatory process was still occurring in the BC group (*p* < 0.001) (Figure 5 and Table 1). At 30 and 60 days, it was observed that the membrane was not completely degraded, suggesting that the inflammatory process was still occurring in all the experimental periods (Figure 6).

#### 3.2.2. Newly Formed Bone

After 7 and 15 days of bone repair, there were similar values in the area of bone neoformation when comparing the groups (Figure 7). After 30 days of bone repair, the group with the collagen membrane presented the greatest values of newly formed bone in relation to the other groups (*p* < 0.001). A large area of bone neoformation was observed in the BC group as opposed to that in the CG group, but the difference was not statistically significant (*p* < 0.05; Figure 7). Regarding the analyzed periods, it was observed that the BG alone showed better bone repair at 30 days than in the other periods (Figure 7).

### 3.3. Immunohistochemical Analysis


*Osteocalcin*


**BG:** Photomicrographs of the repaired bone on day 15 of bone repair showed mild (1) labeling for OC, especially in the extracellular matrix region. At 30 and 60 days of bone repair, it was possible to observe intense marking (3) for the biomarker OC in the region of the bone stump and the center of the defect (Figure 8).

**BC:** The photomicrographs obtained at 15, 30, and 60 days of bone repair showed mild marking (1) for OC (Figure 8).

For semi-quantitative comparison, at 15 days, BG and BC groups showed similar staining areas (*p* < 0.05). At 30 and 60 days, BG levels increased significantly in comparison to the BC group (*p* < 0.05). 


*Osteopontin*


**BG:** Photomicrographs of the repaired bone at 15 days showed moderate (2) labeling for OP. At 30 days, light marking (1) was observed for the biomarker. At 60 days, mild (1) presence of this biomarker was observed again (Figure 9).

**BC:** Regarding OP, mild (1) immunostaining was observed at 15 days of repair, moderate (2) at 30 days, and intense labeling (3) at 60 days of repair (Figure 9).

For semi-quantitative comparison, at 15 and 30 days, there was no difference between BG and BC (*p* > 0.05), whereas, at 60 days, BC showed greater staining than BG (*p* < 0.05). 

## 4. Discussion

This study aimed to evaluate the efficacy of BC membranes (Innovatec’s, São Carlos SP, Brazil), synthesized through bacterial cellulose, in the repair of 8-mm bone defects generated in the skulls of adult male rats. Concerning the membrane’s osteopromotive features and neoformation capacity, the photomicrographs taken after extended periods of bone repair (30 and 60 days) showed no filling of the bone defect with neoformed bone. In addition to the absence of bone tissue, the predominance of fibrous connective tissue throughout the area of the bone defect was noted, even in the later periods. In the initial periods (7 and 15 days), the microscopic characteristics showed, specifically in the BC group, extensive inflammatory infiltrate, which delayed bone repair as opposed to the BG group (Bio-Gide^®^).

The investigation of bone tissue biology as a function of the BC membrane in this study was based on tissue engineering, which has shown promising results, mainly with respect to soft-tissue repairs, such as in the repair of venous ulcers [18,22], traumatic lacerations and abrasions, venous stasis, skin graft donor sites, lesions due to second-degree burns, and ulcers in diabetic feet. Despite evidence of good behavior in soft-tissue repair, little is known about the effect of BC on hard tissues. Therefore, it is necessary to evaluate its efficacy in GBR because it is a natural and biocompatible material [38]. Some studies [35,39] on this subject have obtained promising results that differed from the results found in this study.

To evaluate the osteopromotive property, the critical size defect, which does not heal spontaneously, was chosen because this is the only means by which the biological performance of the biomaterials can be observed, given that the CG group was not able to close the defect [40]. Based on the principle of GBR, the presence of a membrane, as a barrier to not allow the invagination of epithelial cells to the defect area, is essential. Therefore, the membranes used for this purpose must be able to maintain the volume and be biocompatible. 

Porcine collagen membranes are widely used for this purpose, with Bio-Gide^®^ being the most used. Several clinical [41] and experimental studies [42] have demonstrated the effectiveness of this membrane as a biological barrier in significant bone defects, whether associated with a biomaterial or used in isolation.

In the current study, the histometric analysis showed that Bio-Gide^®^ was significantly superior to the BC membrane in terms of the amount of bone formed (*p* < 0.001) after 30 days of bone repair. Histological analysis revealed that despite filling the created bone defect, the BC membrane could not produce mature bone tissue in which it was possible to observe connective tissue alone, with a discrete presence of the membrane and bone tissue.

In contrast, Lee et al. [41] compared the effectiveness of the BC membrane and collagen membrane (GENOSS, Suwon, South Korea) as biological barriers in 8-mm defects in rat skulls. However, in addition to the membrane, a xenogenous bone graft was used under the defect. The authors observed that the bacterial cellulose membrane showed behavior very similar to that of the collagen membrane, where it could stimulate adequate bone neoformation in the created defect and could be used as a biological barrier.

Although the authors reported good results with the BC membrane, it is important to note that an associated xenogenous bone graft was used, and the collagen membrane was not Bio-Gide. However, the results showed the need to carry out further studies on the use of BC membranes in GBR (use of membrane-covered bone substitutes) since the material has very interesting properties, besides being phytotherapeutic. In addition, the reparative cellular activity of this group as a function of immunoblotting for the proteins OP and OC at 60 days was intense and mild, respectively, throughout the area of immunostaining in the extracellular matrix. These proteins represent the maturation of bone tissue. In this way, the use of BC membranes showed interesting cellular activity, indicating the need to evaluate its behavior as a bone substitute in future studies.

The inflammatory activity observed in the histological analysis, which persisted in almost all periods, may be attributed to the fact that cellulose presents itself as a foreign substance in the human body since it cannot be digested, which may have led to a delay in its reabsorption, leading to a greater inflammatory response in the bone tissue (as observed in Figure 5 and Figure 6 and Table 1). 

Thus, despite the limitations of this animal study, BC showed interesting properties as a biological membrane and presented excellent results in the repair of large soft-tissue injuries. It did not demonstrate acceptable results in critical defects in rat skulls when compared to the collagen membrane (Bio-Gide^®^—Geistlich Wolhusen, Switzerland), which proved to be significantly superior in the quality and quantity of neoformed bone than the BC membrane. To improve the performance of the BC membrane, some strategies, like enzyme embedding, to improve the biodegradable characteristic, protein incorporation, to increase the osteopromotive cell grip, could be used, with the proposition that the BC membrane could present better results related to new bone formation [43].

## 5. Conclusions

The results support the conclusion that the membranes we examined had different biological behaviors in the bone neoformation process. The positive control group (BG group) showed the best results, as expected. The test group (BC group) showed low biocompatibility, with a large amount of mature connective tissue in the final stage.

## Figures and Tables

**Figure 1 membranes-10-00230-f001:**
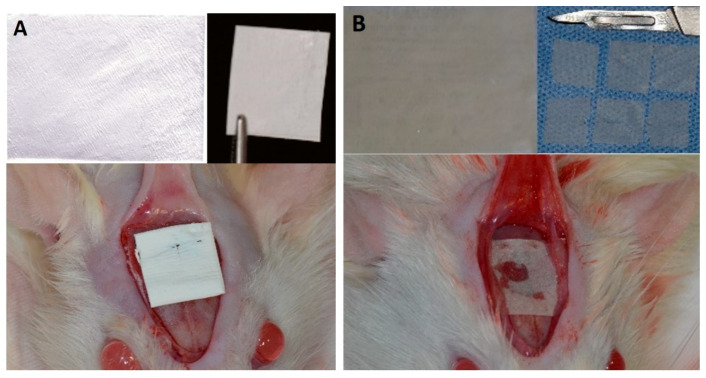
(**A**) Porcine collagen membrane and (**B**) Bacterial cellulose membrane, represented on a microscopical aspect and in the clinical experimental surgery.

**Figure 2 membranes-10-00230-f002:**
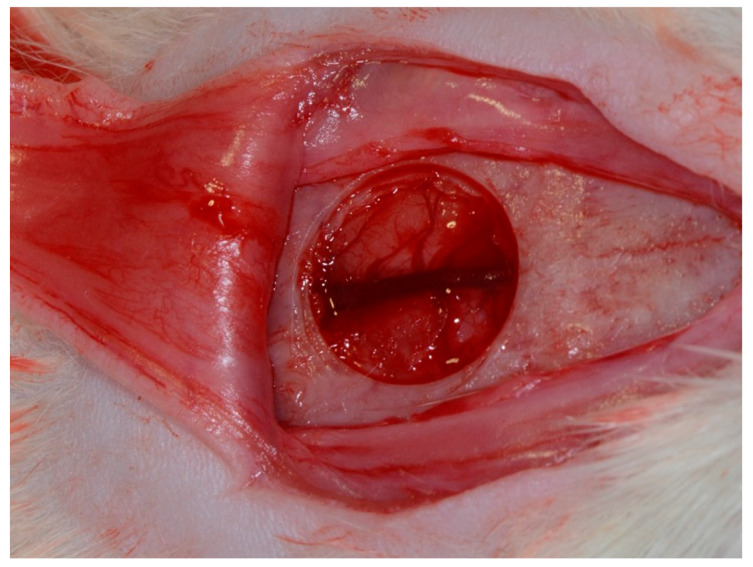
Surgical approach to calvaria and defect creating.

**Figure 3 membranes-10-00230-f003:**
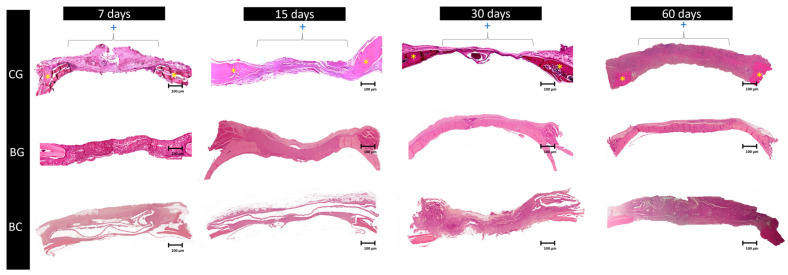
Photomicrographs of the least magnified histological sections (×6.3) for the experimental groups (CG: clot group; BG: Bio-Guide^®^; BC: bacterial cellulose) in all periods analyzed (postoperative 7, 15, 30, and 60 days) in which we observed the bone repair capacity of each membrane. The Bio-Guide^®^ performed better at 60 days with significant closure of the bone defect. The BC group showed good closure of the defect, but with a high prevalence of poorly differentiated connective tissue. * represents regions of the bone stumps; + represents the horizontal extension of the bone defect.

**Figure 4 membranes-10-00230-f004:**
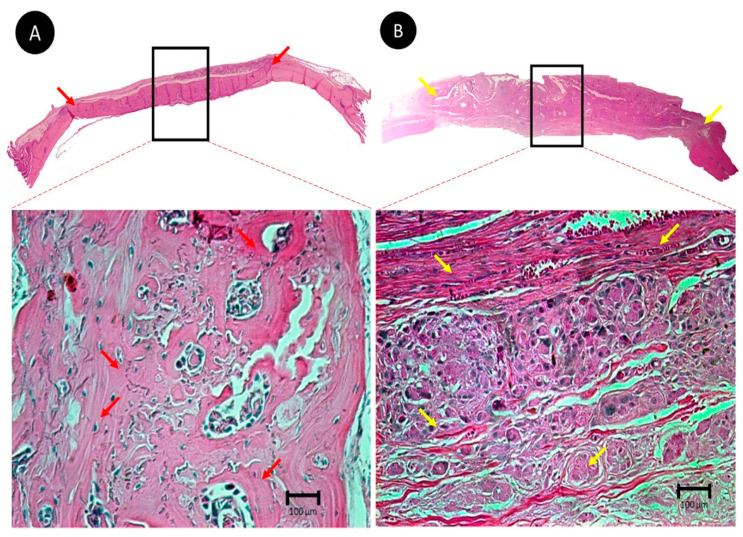
Photomicrographs of the smallest (×6.3) and largest (×25.0) histological sections of the delimited area (center of the defect) in the BG and BC experimental groups in the longest repair period (60 days). (**A**) The BG group; note the presence of bone neoformation from bone stumps and in the center of the defect (red arrows). (**B**) The BC group with a large amount of mature connective tissue and the presence of inflammatory infiltrate (yellow arrows).

**Figure 5 membranes-10-00230-f005:**
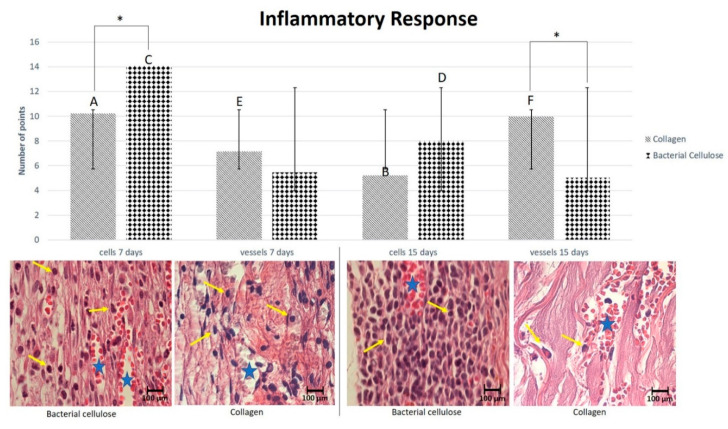
Graph comparing the average numbers of inflammatory cells (yellow arrows) and vessels (blue stars) between the groups and periods analyzed (×100). Regarding the postoperative periods of 7 and 15 days, there was a decrease in the number of inflammatory cells (*p* < 0.001) in the BC and BG groups; there was a statistically significant difference in the numbers of vessels between the BG and BC groups (*p* < 0.001). (*) denotes intergroup statistical difference; capital letters denote the intragroup statistical difference. In short, for inflammatory cells: BC-7 days > BG-7 days > BC-15 days > BG-15 days (*p* < 0.05); for vessels: BG-15 days > BG-7 days > BC-7 days > BC-15 days (*p* < 0.05).

**Figure 6 membranes-10-00230-f006:**
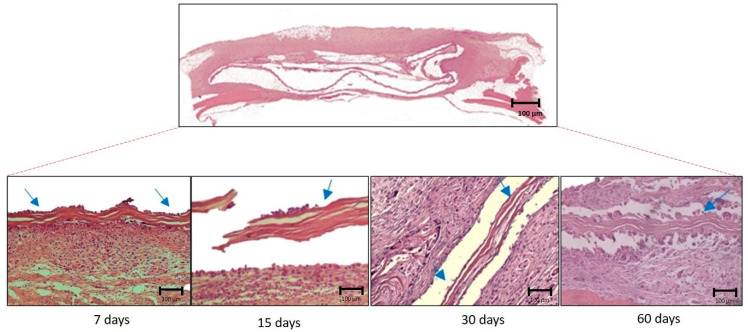
Photomicrographs of the smallest (×6.3) and largest (×12.5) histological sections for the BC experimental group in all postoperative periods (7, 15, 30, and 60 days), demonstrating the presence of the bacterial cellulose membrane (blue arrows).

**Figure 7 membranes-10-00230-f007:**
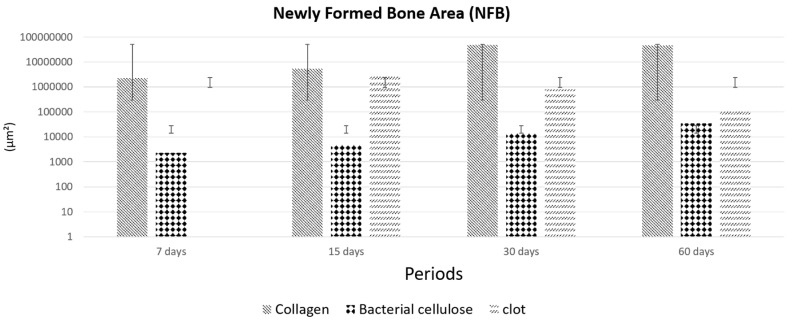
Graph comparing the areas of bone formation between the groups and periods analyzed. When comparing the type of membrane used, there was a statistical difference only in the 30 and 60-day periods between the BG × BC and BG × CG groups (Tukey’s test, *p* < 0.001). A comparison of the analyzed periods showed a statistical difference only in the BG group between the periods 30 d and 60 d × 15 d and 30 d and 60 d × 7 d (*p* < 0.001). In short, at 7 days: BG > BC > C; 15 days: BG > C > BC; 30 days: BG > C > BC; 60 days: BG > C > BC (*p* < 0.05).

**Figure 8 membranes-10-00230-f008:**
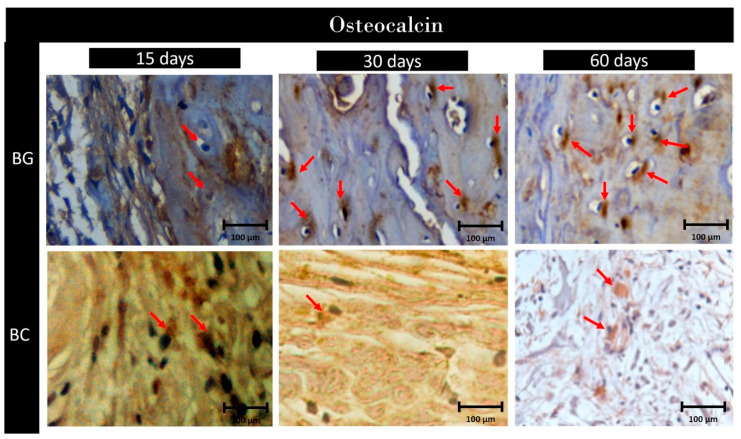
Photomicrographs of the immunohistochemical analyses in 15, 30, and 60 days highlight osteocalcin reaction at ×40.0 magnification. The red arrows indicate labeling. 1, 2, and 3 indicate mild, moderate, and intense labeling, respectively.

**Figure 9 membranes-10-00230-f009:**
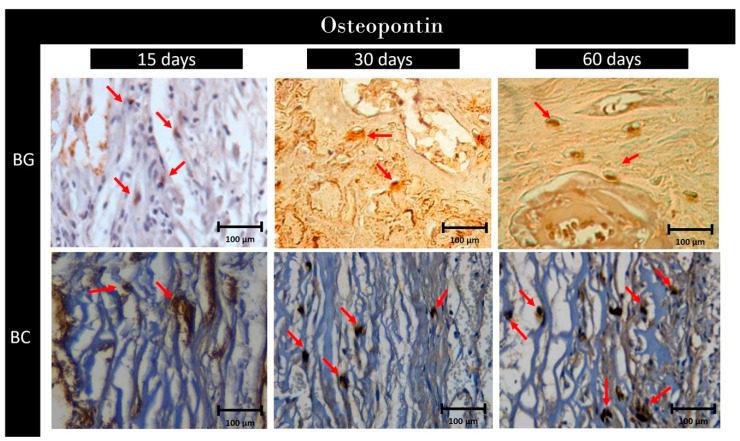
Photomicrographs of the immunohistochemical analyses at postoperative days 15, 30, and 60 highlight osteopontin reaction at ×40.0 of magnification. The red arrows denote labeling. 1, 2, and 3 indicate mild, moderate, and intense labeling, respectively.

**Table 1 membranes-10-00230-t001:** Table demonstrating the average and standard deviation for inflammatory cells and vessels for each group (BG and BC) at postoperative 7 and 15 days. *p*-values for intergroup comparison.

Membranes	7		15	
	Cells	Vessels	Cells	Vessels
Collagen	10.22 ± 2.08 *	7.14 ± 1.05	5.20 ± 2.40	9.98 ± 2.22
Bacterial	14.06 ± 3.11	5.46 ± 2.33	8.02 ± 1.25	5.04 ± 1.29
	*p* = 0.019	*p* = 0.163	*p* = 0.072	*p* < 0.001

* Average; (standard deviation)

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
