# Peer review of "Is the Bacterial Cellulose Membrane Feasible for Osteopromotive Property?"

_membranes, 2020, doi:10.3390/membranes10090230_

Round 1
Reviewer 1 Report
The manuscript entitled 'Is the bacterial cellulose membrane feasible for osteopromotive property?' by Bassi et al. presents a comparative analyses on the effectiveness of different membranes which shall be used as a mesh for fillilling critical size defects with blood clots in order to promote bone regeneration. The whole set of analyses is fully and solely based on semi-quantitative histologic investigations. Although the topic is of high clinical importance, a bunch of concerns basically addressing the readout parameters deem this manuscript unable for publication as it is.
Further experiments are needed and re-analyses of histological sections have to be performed:
Major concerns:
- Numbers of sections analysed per animal have to be provided in order to estimate reliability of results
- To clarify vessel staining a staining for endomucin and CD31 should be conducted. HE staining is not sufficient.
- In this way, also immune cell staining at least for CD45 should be presented to ensure reliability of results.
- To evaluate bone formation, µCT and von Kossa staining have to be conducted and presented.
- Fibrotic and necrotic areas should be determined (e.g. Tunel)
- Data on biocompatibility of membranes in rats should be provided (e.g. by signs of systemic inflammation such as CRP or in vitro using a whole blood assay)
Minor concerns:
- Title has to be revised clearly stating what was compared mentioning the superior membrane.
- Statistics must be clarified for each assessment in the figure legend
- Quantification of OC and OP is missing
- Graphs have to be revised e.g. „points“ is not a proper axis title
- Arrive guidelines have to be followed and methods need to be introduced
- Husbandry and housing have to be included
- Staining background of OC and OPN varies tremendously which needs to be explained. Of note, OC and OPN stainings should stain ECM and not the cells...??? Please explain. Furthermore, OPN is not a late marker of osteogenic differentiation and can also be expressed by a variety of immune cells, which might be stained here instead!
- ImageJ provides a powerful tool to perform histological analyses, which should be used instead of scoring
Author Response
Dear Reviewer, all suggestions and comments were carefully studied and answered in the attached document.
Best Regards

Reviewer 2 Report
The paper by Ana Paula Farnezi Bassi et al. entitled ‘Is the bacterial cellulose membrane feasible for osteopromotive property?’ evaluates bacterial cellulose membrane in the process of bone regeneration. Although the cellulose membrane was not capable of stimulating bone neoformation, the research is well described and this ‘negative result’ is worth publishing.
Comments and Suggestions for Authors
There are some points that should be addressed:
- Figure 1.
If possible, image showing more detail structure of the materials would be valuable, with a scale and brief desription of the method picture was taken by.
- Sections 2.4. Histological Analysis and 2.5. Histometric Analysis
This quite a long part is doubled, contained in both sections:
'The photomicrography of the blades was made using an optical microscope (LeicaR DMLB, Heerbrugg, Switzerland) coupled to an image capturing camera (LeicaR DC 300F microsystems ltd, Heerbrugg, Switzerland) and connected to a microcomputer with ImageJ digitized image analyzer software (National Institutes of Health, Bethesda, MD, USA).'
Maybe these two sections could be joined together 'Histological and Histometric Analysis' to avoid repeating of the instruments used.
- Lines 185-188
The definition of the scores of immunohistochemistry could be expressed as a range?
e.g. score 2 is 25%-50%
I think '15% of stained cells' is up to 25% as well as up to 50% as well as up to 75% ....
- Page with Figure 3 rotated to 'landscape' orientation corrupted the numbering of pages.
- Figure 4 legend
I do not understand ' the smallest (×6.3) and largest (×25.0) histological sections' It is magnification?
It would be for sake of the reader to clearly state that (A) is the BG group.
- Lines 241 and 246
Please check the wording. 'have noticed during the analysis were' ' BC groups, for, there was'
- Table 1
What about rearranging the data in the following way:
First line: 'cells' and 'vessels'
Third line: 'average' 'standard deviation' 'average' 'standard deviation'
And proper rearrangement of the values...
- Figure 6
' the smallest (×6.3) and largest (×25.0) histological sections' ?
The scale bars are strange, it cannot be of the same length for lower and higher magnification.
BG and BC experimental groups? I think it is only BC group.
- Figure 5 and 7
The lines showing probably standard deviation are the same for all graphs.
Lines 297 -308 Check if it is correct and whether the text is compatible with the graphs in the Figure 7.
- Figure 8 and 9
'1, 2, and 3 indicate mild, moderate, and intense labeling, respectively.'
It is not in the images...
- Lines 348-361
This part is more appropriate for the introduction. At least the explanation of the 'critical size defect'.
- Format of the references is very heterogeneous.
Author Response
Dear revisors, thank you for your comments, the team made the corrections that are explained in the attached document.

Reviewer 3 Report
Thank you for giving me this opportunity to review this interesting in vivo study entitled, "Is the bacterial cellulose membrane feasible for osteopromotive property?". I here carefully reviewed the submitted set of the manuscript and found it merits of publication. However there need some revisions as suggested below to meet the scientific standard for publication I'm afraid.
- In the introduction section the properties and characteristics of BC should be well more described in details referring to recent publications in oral and maxillofacial surgery, such as Materials (Basel) 2019 Aug; 12(15): 2489. Published online 2019 Aug 6. doi: 10.3390/ma12152489 and Acta Biomaterialia Volume 9, Issue 4, April 2013, Pages 6116-6122.
- BC can't obtain a bioresorbable property, and then have to be removed. this shortcoming has to be well introduced and discussed.
- As the results obtained this study, BC might need more modifications for bone regenerative promotion at a critical bone defect in maxillofacial application. The authors should need to add more strategies and discuss these in details in the discussion section.
Author Response
Dear revisors, thank you for your comments, the team made the corrections that are explained in the attached document

Reviewer 4 Report
In this paper, the authors evaluated the possibility of using bacterial cellulose membrane as a scaffold to promote bone regeneration. In this paper, the authors used bacterial cellulose membrane as an alternative scaffold to replace the currently used collagen membrane, but I cannot understand why the research was conducted with this membrane. The value of this paper cannot be appreciated because the findings and facts that this membrane is not suitable for promoting bone regeneration do not convey enough new information to this journal.
Author Response
Dear revisor, thank you for your comments, the team made the corrections that are explained in the attached document

Round 2
Reviewer 3 Report
Thank you for letting me re-review this revised manuscript.
I here carefully re-reviewed the submitted set of the manuscript and found it merits of publication.
Author Response
Dear revisor, thank you for your previously considerations.
This manuscript is a resubmission of an earlier submission. The following is a list of the peer review reports and author responses from that submission.
Round 1
Reviewer 1 Report
The manuscript addresses a potentially interesting issue concerning the possibility of using bacterial cellulose membrane for osteopromotive purposes.
In general, the paper is written within the Journal required style. However, there are some important concerns about experimental protocol, especially related to the stages of bacterial cellulose synthesis and its characterization. I am afraid, that considering the information given by the authors, it would not be possible to repeat the experiments carried out. Moreover, based on the descriptions of the procedures for the analyzes performed it is difficult to be sure that the experiments were conducted in the way which allow to obtain correct results and carry out its proper interpretations. The exact description of this information is particularly important considering the authors' conclusion: “It concludes that despite the BC membrane showing a promising application in the soft tissue repair, it did not allow bone repair in rats calvaria.” This conclusion clearly proves the unsuitability of bacterial cellulose in orthopedic applications. However, in my opinion, such observation may result from improperly carried out cellulose synthesis process or the subsequent stages of its preparation. The authors do not seem to be specialists in research related to the bacterial cellulose. They make such fundamental mistakes as confusing the microorganisms that produce this biomaterial: line 63: “In this context, a cellulosic membrane produced by the action of some yeast species on green tea (Nanoskin® ) …”. Yeast species (even on green tea – which is also untypical) do not produce cellulose. In conterast, in the Materials and Methods section, the Authors report that cellulose was produced by bacteria (Gluconacetobacter xylinus). Regarding the bacteria, G. xylinus is already the outdated name of the bacterium, which has been classified as Komagataeibacter xylinus for several years.
There is no information on the method used for the production of cellulose. It is not known under what conditions the process was carried out and for how long. How and whether the obtained biomaterial was purified from cells and medium residues. The above conditions affect the properties of the obtained bacterial cellulose, which may be of significant importance in the context of the tests performed. How the K. xylinus cells were obtained for cellulose production. Why was green tea as a nitrogen source used, what concentration was used? This is not a typical ingredient in cellulose production medium. There are many different types of green tea, which one the authors used, how it influenced the properties of the material. How is it possible that the bacteria survived the autoclave process? – line 80: “The bacteria were then inoculated in the culture medium and, after being added, the medium was autoclaved at 100 °C.” What is the autoclave process that is carried out at 100°C? On what basis the authors believe that cellulose was produced by a specific bacterial species (isolation and identification is missing). It is impossible, that the cellulose “extracted from the culture medium was in a pure 3D structure” - what happened to the cells and components of the medium? Finally, there is no information regarding the properties of the cellulose (cytotoxicity, mechanical properties, purity etc.)
In addition, I did not find any information about cellulose marked by the authors as Nanoskin®. Authors should provide the appropriate citation or website with relevant information about this product.
L291 - writing about a bacterial cellulose membrane as a green tea membrane is inappropriate.
Considering the above remarks, in my opinion, the conclusions drawn by the authors, based on uncertain results and the use of uncharacterized material, could be dangerous for the further development of science related to the possibilities of cellulose application in bone regeneration processes.
Author Response
Dear reviewer, thank you for your valuable comments, the authors have made the corrections which are explained below.
Reviewer 1
- In general, the paper is written within the Journal’s required style. However, there are some important concerns about the experimental protocol, particularly related to the stages of bacterial cellulose synthesis and its characterization. I am afraid that considering the information given by the authors, it would not be possible to repeat the experiments carried out. Moreover, based on the descriptions of the procedures for the analyzes performed it is difficult to be sure that the experiments were conducted in the way which allows to obtain correct results and carry out its proper interpretations.
- We would like to make it clear to the reviewer that this is a membrane developed in Brazil by Innovatec-Produtos Biotecnológicos Ltda. (São Carlos SP, Brazil) with the certification; FIESP: 80591940001, ISO: 13485:2003/AC:2009 (lines 88 to 90). Therefore, this is a membrane that any individual can purchase. In particular, this membrane was provided for an ‘in vivo’ study to analyse bone tissue repair, since, in soft tissue, the same membrane has been showing good clinical performance. This interest arose from a single article published with a bone tissue repair process that obtained an opposite result from the present study. (Lee SH, Lim YM, Jeong SI, An SJ, Kang SS, Jeong CM, Huh JB. The effect of bacterial cellulose membrane compared with collagen membrane on guided bone regeneration. J Adv Prosthodont 2015;7(6):484-95).
- As for the analysis, there are already based on previous studies that we have been developing within the department for some time to assess the osteopromotive factor of various membranes available on the market (reference 12: Danieletto-Zanna, C.F.; Bizelli, V.F.; Ramires, G.A.D.A.; Francatti, T.M.; de Carvalho P.S.P.; Bassi, A.P.F. Osteopromotion Capacity of Bovine Cortical Membranes in Critical Defects of Rat Calvaria: Histological and Immunohistochemical Analysis. Int J Biomater. Feb 2020, 6426702). Thus, we believe that the analyzes used are adequate since with it we can evaluate both the qualitative and quantitative characteristics of bone neoformation as well as its biological behavior with the use of immunohistochemical markers that allow us to evaluate both the potential of bone neoformation and the maturation of the tissue.
- The exact description of this information is particularly important considering the authors' conclusion: “It concludes that despite the BC membrane showing a promising application in the soft tissue repair, it did not allow bone repair in rats calvaria.” This conclusion clearly proves the unsuitability of bacterial cellulose in orthopedic applications. However, in my opinion, such observation may result from improperly carried out cellulose synthesis process or the subsequent stages of its preparation.
We would like do make it clear that this conclusion was based on the specific findings on this study. At no time do we disqualify the use of this or another cellulose-based membrane for other purposes, not least because this is a specific study on a critical bone defect. As mentioned earlier, we found only one article using the same membrane in the ROG process, however; the authors associated this membrane with the use of biomaterials, which probably facilitated bone regeneration (Lee SH, Lim YM, Jeong SI, An SJ, Kang SS, Jeong CM, Huh JB. The effect of bacterial cellulose membrane compared with collagen membrane on guided bone regeneration. J Adv Prosthodont 2015;7(6):484-95). At this point we reinforce that we are not specialists in cellulose bacterial, but we carried out work analyzing membranes in the ROG process for some years and this is the context and the main objective of this article, at no time do we want to discuss the synthesis/fabrication forms of the membrane, not least because it has gone through all the steps certification and production, which is available for purchase in the market.
- They make such fundamental mistakes as confusing the microorganisms that produce this biomaterial: line 63: “In this context, a cellulosic membrane produced by the action of some yeast species on green tea (Nanoskin® ) …”. Yeast species (even on green tea – which is also untypical) do not produce cellulose. In contrast, within the Materials and Methods section, the authors report that cellulose was produced by bacteria (Gluconacetobacter xylinus). Regarding the bacteria, G. xylinus is already the outdated name of the bacterium which has been classified as Komagataeibacter xylinus for several years. There is no information on the method used for the production of cellulose. It is not known under what conditions the process was carried out and for how long. How and whether the obtained biomaterial was purified from cells and medium residues. The above conditions affect the properties of the obtained bacterial cellulose, which may be of significant importance in the context of the tests performed. How the K. xylinus cells were obtained for cellulose production. Why was green tea as a nitrogen source used, what concentration was used? This is not a typical ingredient in cellulose production medium. There are many different types of green tea, which one of the authors used, how it influenced the properties of the material. How is it possible that the bacteria survived the autoclave process? – line 80: “The bacteria were then inoculated in the culture medium and, after being added, the medium was autoclaved at 100 °C.” What is the autoclave process that is carried out, at 100°C? On what basis the authors believe that cellulose was produced by a specific bacterial species (isolation and identification are missing). It is impossible that the cellulose “extracted from the culture medium was in a pure 3D structure” - what happened to the cells and components of the medium? Finally, there is no information regarding the properties of the cellulose (cytotoxicity, mechanical properties, purity, etc.)
- As for the synthesis part of the membrane, it is not and never was the objective of this study since it comes ready from the factory and it has all the criteria and certifications for its production. The description contained in the materials and methods was provided by the manufacturer itself and therefore; it is not up to us to modify or discuss their formation and manufacturing. This form of membrane synthesis has already been published in the reference 23 (De Olyveira, G.M.; Filho, L.X.; Basmaji, P.; Costa, L.M.M. Bacterial nanocellulose for medicine regenerative. J. Nanotechnol. Eng. Med. 2011, 2, 1-8. Doi:10.1115/1.4004181.) The hemicellulose developed by Innovatec-Produtos Biotecnológicos Ltda. (São Carlos SP, Brazil), is composed of a mixture of bacteria and yeast from the green tea, providing a culture medium in which the biological microorganism Acetobacter xylinum (2), a Gram-negative bacterium, develops the product with biodegradable, biocompatible characteristics, non-toxic and non-allergenic. (Kongruang S. Bacterial cellulose production by Acetobacter xylinum strains from agricultural waste products. Appl Biochem Biotechnol. 2008; 148:245–56.)
- In addition, I did not find any information about cellulose marked by the authors as Nanoskin®. The authors should provide the appropriate citation or website with relevant information about this product.
- We would like, however, to reaffirm that this membrane has several previous studies, and that is why it encouraged us to use it to assess its potential barries in guided bone regeneration processes. References 12, 20, 22, 23, 24 and 25.
- L291 - writing about a bacterial cellulose membrane as a green tea membrane is inappropriate.
- In line 291 the term green tea membrane was corrected to bacterial cellulose membrane.
Reviewer 2 Report
This work presents the efficiency of bacterial cellulose membrane (Nanoskin®) in the bone repair to collagen membrane Bio-Gide® in rat calvaria defects. The paper is well written and organized but the conclusions should be sustained by more results and therefore, it would highly benefit from the following adjustments:
- the introduction should present the topic more widely and could benefit form more citations. For example, the following paper could be relevant:
Codreanu, A.; Balta, C.; Herman, H.; Cotoraci, C.; Mihali, C.V.; Zurbau, N.; Zaharia, C.; Rapa, M.; Stanescu, P.; Radu, I.-C.; Vasile, E.; Lupu, G.; Galateanu, B.; Hermenean, A. Bacterial Cellulose-Modified Polyhydroxyalkanoates Scaffolds Promotes Bone Formation in Critical Size Calvarial Defects in Mice. Materials 2020, 13, 1433.
- the methods could be presented in more detail. Brief protocols should be presentated for each method. If the protocol was previously published or reproduced from another paper, an appropriate citation should be added.
- the experimental design should include the evaluation of some of the osteogenic specific markers. Protein or gene expression of OPN, OCN, etc could be done. Alternatively the levels of ALT could be investigated and at least Alyzarin Red staining should be done. In this view, the following paper could be relevant.
E. Tanasa, C. Zaharia, A. Hudita, I.C. Radu, M. Costache, B. Galateanu,
Impact of the magnetic field on 3T3-E1 preosteoblasts inside SMART silk fibroin-based scaffolds decorated with magnetic nanoparticles,
Materials Science and Engineering: C, Volume 110, 2020, 110714, ISSN 0928-4931, https://doi.org/10.1016/j.msec.2020.110714
Author Response
Dear reviewer, thank you for your valuable comments, the authors have made the corrections which are explained below.
Reviewer 2
- The introduction should present the topic more widely and could benefit from more citations. For example, the following paper could be relevant:
- The paper suggested was read and the collected information was added in lines 69 to 71 (reference 21).
- the methods could be presented in more detail. Brief protocols should be present for each method. If the protocol was previously published or reproduced from another paper, an appropriate citation should be added.
- In line 146, the reference 12 was added to reference the appropriate paper.
- In line 162, the references 12, 26 and 27 were added to reference the appropriate papers.
- the experimental design should include the evaluation of some of the osteogenic specific markers. Protein or gene expression of OPN, OCN, etc could be done. Alternatively, the levels of ALT could be investigated and at least Alyzarin Red staining should be done. In this view, the following paper could be relevant.
- The osteogenic markers OPN (Osteopontin) and OCN (osteocalin) were analyzed at the immunohistochemical technique as explained in lines 164 to 182. The levels of ALT and Alyzarin Red staining could be a great analysis for future studies, because all animals who composed this study have already been euthanized. Those analyses should be possible only through calcified samples, and our study used decalcified bone tissue. However, we appreciate the contribution.
Reviewer 3 Report
In this manuscript, the bone repair effects of both bacterial cellulose (BC) and collagen membrane (BG) were elucidated in vivo model, which is rats. However, I feel like the manuscript is not well written at the current stage, especially in the results and discussion sections. There is room for improvement.
Article is well prepared nevertheless some comments have to be done before publishing.
- Materials and methods comments: Again, sound structuring and grammar enables the reader to appreciate and, if desired, repeat, the experimental protocols.
- Results - Errors/comments: Although presented as a combined Results & Discussion section, the results and associated figures were described in sufficient detail to enable the reader to appreciate the raw data.
- Why did the author use the SEM in Figure 1, to demonstrate the surface morphology? AFM should provide more in-deep information about the ultrastructure of collagen and surface morphology than SEM.
- There is no statistical analysis of H&E stain included in the inflammatory cells and vessels described in Figures 5 and Table 1. Furthermore, the normalization should be better described. I would suggest the author to provide the many area datas of the inflammatory cells and vessels studies as a supplementary information.
- BC is not biodegradable in the human body because it does not have the cellulose degrading enzyme (cellulase). The degradation and swelling ratio profile of the BC also needs to be demonstrated.
- Discussion - Errors/comments: I feel that the lack of a distinct discussion is the relative weakness of this manuscript
- English style, grammar and lexicon should be revised in the abstract and in the main text. After the proper amendments, I strongly encourage re-submission.
-References add
(1)Int. J. Mol. Sci. 2017, 18(11), 2236; https://doi.org/10.3390/ijms18112236
(2)Biotechnology and Bioprocess Engineering volume 20, pages948–955(2015)
Author Response
Dear reviewer, thank you for your valuable comments, the authors have made the corrections which are explained below.
Reviewer 3
- Materials and methods comments: Again, sound structuring and grammar enables the reader to appreciate and, if desired, repeat, the experimental protocols
- Dear reviewer, thank you a lot for the comment.
- Results - Errors/comments: Although presented as a combined Results & Discussion section, the results and associated figures were described in sufficient detail to enable the reader to appreciate the raw data.
- Dear reviewer, thank you a lot for the comment.
- Why did the author use the SEM in Figure 1, to demonstrate the surface morphology? AFM should provide more in-deep information about the ultrastructure of collagen and surface morphology than SEM.
- In figure 1, was not used SEM to demonstrate the surface morphology, the picture was made by a semi-professional photographic camera (CANON) only to demonstrate the membrane structure. AFM, for sure could provide great information about the ultrastructure of collagen and surface morphology but, we don´t have enough resources to realize this analysis, especially at this moment in our country, which outbreaks due to COVID 19 and it still maintains a great number of cases, and then, all labs are closed, not allowing other specific analysis.
- There is no statistical analysis of the H&E stain included in the inflammatory cells and vessels described in Figures 5 and Table 1. Furthermore, the normalization should be better described. I would suggest the author to provide the area data of the inflammatory cells and vessels studies as supplementary information.
- The statistical analysis of H&E stain in figure 5 and Table 1 has added as recommended. Lines 233 and 236.
- BC is not biodegradable in the human body because it does not have the cellulose degrading enzyme (cellulase). The degradation and swelling ratio profile of the BC also needs to be demonstrated.
- The degradation and swelling ration profile of the BC were demonstrated at figure 6 as recommended. Line 238 - 241.
- - English style, grammar and lexicon should be revised in the abstract and in the main text. After the proper amendments, I strongly encourage re-submission.
- The manuscript was submitted to a specialized publisher that ensured the correction of the English style, grammar and lexicon. Follow below the letter of English native revision:

Round 2
Reviewer 2 Report
The manuscript was improved and I consider it is appropriate now for publication.
Author Response
Dear reviewer, thank you for the attention

Reviewer 3 Report
Accept in present form
Author Response

(The authors gave the same response as above.)
